# Extracting Chinese events with a joint label space model

**Wenzhi Huang[1,2], Junchi Zhang[2], Donghong Ji[1] ***

**1** Key Laboratory of Aerospace Information Security and Trusted Computing, Ministry of Education, School of Cyber Science and Engineering, Wuhan University, Wuhan, China, **2** School of Computer Science and Engineering, Wuhan Institute of Technology, Wuhan, Hubei, China

* dhji@whu.edu.cn

**Data Availability Statement:** Data is available at: https://github.com/zjcerwin/cnlabelattn.

**Funding:** This work is supported by the Natural Science Foundation of China (No. 62106179).

## Abstract

The task of event extraction consists of three subtasks namely entity recognition, trigger identification and argument role classification. Recent work tackles these subtasks jointly with the method of multi-task learning for better extraction performance. Despite being effective, existing attempts typically treat labels of event subtasks as uninformative and independent one-hot vectors, ignoring the potential loss of useful label information, thereby making it difficult for these models to incorporate interactive features on the label level. In this paper, we propose a joint label space framework to improve Chinese event extraction. Specifically, the model converts labels of all subtasks into a dense matrix, giving each Chinese character a shared label distribution via an incrementally refined attention mechanism. Then the learned label embeddings are also used as the weight of the output layer for each subtask, hence adjusted along with model training. In addition, we incorporate the word lexicon into the character representation in a soft probabilistic manner, hence alleviating the impact of word segmentation errors. Extensive experiments on Chinese and English benchmarks demonstrate that our model outperforms state-of-the-art methods.

## Introduction

Event extraction is a field of study that aims to generate structural knowledge with regard to particular occurred events that people care about from plain texts [1, 2]. End-to-end event extraction contains three fundamental tasks, namely entity recognition, event trigger identification and argument role classification. Entities, referred to a set of world objects (e.g. Steve Jobs, Bill Gates), consist of several consecutive tokens in the sentence with an association of a particular type (e.g. Persons, Organizations and Locations). Event triggers, generally determined by verbs or nominalizations, are keywords that can mostly evoke the corresponding events. For example, given a Chinese text:

"军警两名士兵丧生。"(Two soldiers of the military police were killed.)

In this instance, an event detection system should be able to identify that the word "丧生"(were killed) is an event trigger of type *Die*. At last, event arguments are entities to be

**Competing interests:** The authors have declared that no competing interests exist.

connected to triggers with specific roles in the event, such as "士兵"(soldiers) plays an *Victim* role in the *Die* event triggered by "丧生"(were killed).

Traditional pipelined extraction systems treat entity, trigger and argument extractions as three separate tasks, which follow a procedure of entity recognition → trigger word identification → argument role classification [3–8]. Although these methods are flexible, they have the limitations that incorrect entity and trigger results would degrade the performance of argument role classification. These pipelined methods could lead to two issues:1) previous step errors would propagate to following steps;2)they are typically insufficient for modeling the mutual dependence among subtasks. Therefore, later approaches put more focus on building joint models to simultaneously extract entities, triggers and argument roles. Prior joint learning methods depend heavily on human-designed indicator features and pre-built syntax tools to capture most useful information for event extraction [9–11]. With the raise of deep learning models, recent studies concentrate on representation-based neural networks to automatically compose low-dimensional features, and multi-task approaches based on hard parameter sharing are applied to jointly solve information extraction tasks [12–15]. As shown in Fig 1(a), their approaches can be mainly divided into three components: (1) An embedding look-up layer with pre-tokenized words as inputs, the embedding table is usually initialized with pre-

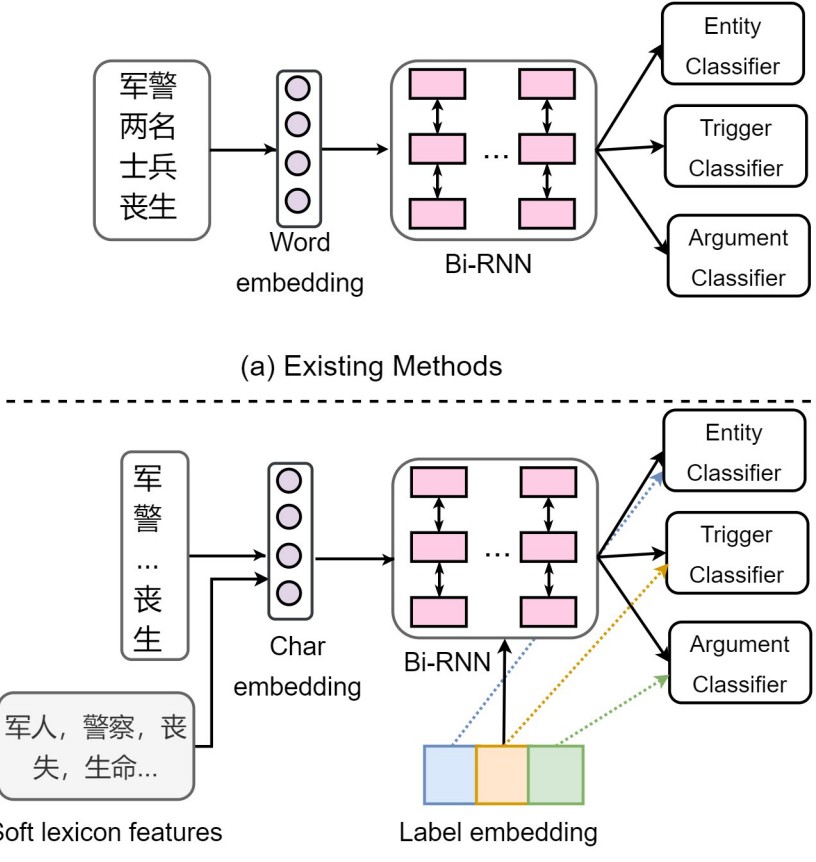

**Fig 1. Illustration of comparison of existing methods and our proposed method for an input Chinese sentence "军警两名士兵丧生(Two military soldiers were killed)".**

trained word vectors [16]; (2) A shared multi-layer Bi-directional recurrent neural network (BiRNN) to encode deep contextual representations, and the long-short term memory alternative (LSTM) [17] is typically adopted for handling the vanish gradient problem. (3) Independent output networks with the *Softmax* function are added on top for classifying specific labels.

Although these neural-based joint learning methods perform better than the former, the complicated event structure still poses two challenges when applied to Chinese event extraction. **First**, unlike languages with explicit word boundaries (e.g. English), Chinese event extraction is more difficult since words in Chinese texts are not indicated naturally. Hence, a word segmentation procedure is often required before involving subsequent applications [18–20]. However, it is unavoidable that words are segmented incorrectly. This will result in inherent errors in the detection of entity and trigger boundaries and the prediction of their categories. Therefore, some approaches resort to performing Chinese event detection directly at the character level [21, 22]. This results in a dilemma between the choice of performing Chinese event extraction based on a fully character-level model and by first segment text into words. **Second**, traditional multi-task models that are based on hard parameter sharing rely on implicit network weights to capture correlations among tasks [13, 23–26], treating event labels as meaningless and independent one-hot vectors, which cause a loss of potential label information. However, this is inconsistent with the process of the human annotation of an event mention. For instance, for a trigger with event type e.g. *Divorce*, a human will only connect PERSON entities to the trigger as argument roles based on the fact that it is impossible for non-human beings to divorce.

Previously, we have presented a transition-based method [27] that approaches the joint learning in a left-to-right decoding order, which has been proven to be better than simple shared-private models. However, it suffers from two limitations: 1) The elaborate modification to the standard LSTM hinders the computation of multiple sentence in a batch and not all lexicons that related to a character are used; 2) The interactive semantics of all task labels have not been fully explored, in the sense that the event label information has not been introduced into the shared representations.

In this work, we introduce a novel **M**ulti-layer **L**abel **A**ttentive framework to improve Chinese **E**vent **E**xtraction (MLAEE). For the above first issue, there have been studies showing that integrating lexicon features into character-based networks could lead to better entity recognition performance [28, 29]. Inspired by these methods, we propose to perform event extraction based on characters and enhance character representations by introducing the word lexicon, which is presented in Fig 1(b). In contrast to modifying LSTM interior to incorporate word embeddings in hidden layers [28], we propose to make use of a simple and effective method [30] that turns the lexicon matching results to the BMES encoding scheme, which bypass the need for a complicated model architecture. For the second issue, we propose a joint label space for all the entity and event types, thereby allowing correlated-type information to incorporate into the network representations. In particular, we map labels of each subtask into low-dimensional semantic vectors, similar to word embeddings [16]. By stacking all label types of events as a matrix, we let each character hidden state performs attention over it for deriving a label importance distribution, and share the label parameters with the output layers. By doing this, label embeddings can be viewed as a semantic bridge that enables interactions between the encoding and decoding stage, leading to a novel joint learning approach.

We conduct sets of experiments on a standard benchmark dataset for event extraction. With comprehensive comparisons with existing advanced methods, our model achieves state-of-the-art results on the Chinese ACE2005 dataset. To demonstrate that our joint label space model is applicable across different languages and tasks, we make two additional experiments:

1)event extraction on the English ACE2005 dataset; 2)using entity relation extraction as an auxiliary task to boost event performance. Results show that our approach is also effective on both the English dataset and incorporating relation labels. Furthermore, we make an ablation analysis to show the contribution of each proposed module and visualization results indicate that label embeddings can indeed capture semantic correlations among entities, triggers and argument roles.

## Task definition

Formally, given an input sentence represented as a sequence of characters $C = c_1, c_2, \ldots c_n$, we extract a set of entities $E$, event triggers $T$ and event arguments $A$. In particular, each token $c_i$ will be determined to be an entity span $e_i$. Then $c_i$ will be differentiated to be a part of a positive or negative trigger word and will be further categorized to an event subtype label $t_i$ if $c_i$ is a positive trigger word. Further, for each trigger $t_i$ and entity $e_i$ pair in the same sentence, an argument role $a_{ij}$ is required to be predicted. Following [9, 13, 25], we prepare argument candidates using predicted entity mentions.

## Methodology

In this section, we will detail our proposed MLAEE model. As Fig 2 illustrates, MLAEE extract event outputs from an input Chinese text in three steps: input representation layer, label attentive encoding layer and event identification layer.

During event decoding, we use two separate sequence taggers to obtain entity and event trigger results, respectively. Then for each entity-trigger pair, we assign it with an event relationship under the definition of argument role types designated by ACE2005 https://catalog.ldc.upenn.edu/ldc2006t06. Note that the weights of entity, trigger and argument role output networks are stacked as one label embedding matrix, which will be used at the encoding layer.

## Input embedding

At input embedding layer, a hybrid approach that both character-level and word-level features are used for input representations. In particular, for an input Chinese sequence consisting of $n$ characters $C = c_1, c_2, \ldots c_n$, we transform the one-hot vectors into the distributed representations $E^c = e_1^c, e_2^c, \ldots e_n^c$ with a deep transformer layer. Its weight is pre-trained on a large amount of raw text with the object of the masked bi-directional language model [31]. To be consistent with BERT pre-training, we add two special tokens [CLS] and [SEP] at the front and end of $C$, respectively, before obtaining $E^c$. These contextualized embeddings have been proved to be better than static word embeddings [31], e.g. Word2Vec, Glove in many natural language tasks, due to dynamic embeddings are more similar to the diverse nature in human languages in the sense that the meaning of a word is changed along with its surrounding words.

## Soft lexicon features

A flaw of the merely character-based event extraction approach is that the word information can not be utilized correspondingly. In this work, we investigate a *SoftLexicon* approach [30] that simply augments current character representation $c_i$ with all matching word embeddings in a soft probabilistic manner. In particular, as presented in Fig 2 (bottom right), *SoftLexicon* first extract all words that contain character $c_i$ in a lexicon and only keep the words that can be found in the input sequence. Then based on the location of $c_i$ in a matched word $w_i$, which can be in the **B**egin, **M**iddle, **E**nd of $w_i$ or a **S**ingle-character word, $w_i$ is categorized and marked as one of the four segmentation labels $K = \{\mathbf{B}, \mathbf{M}, \mathbf{E}, \mathbf{S}\}$. For example, the character $c_7$ ("丧") in

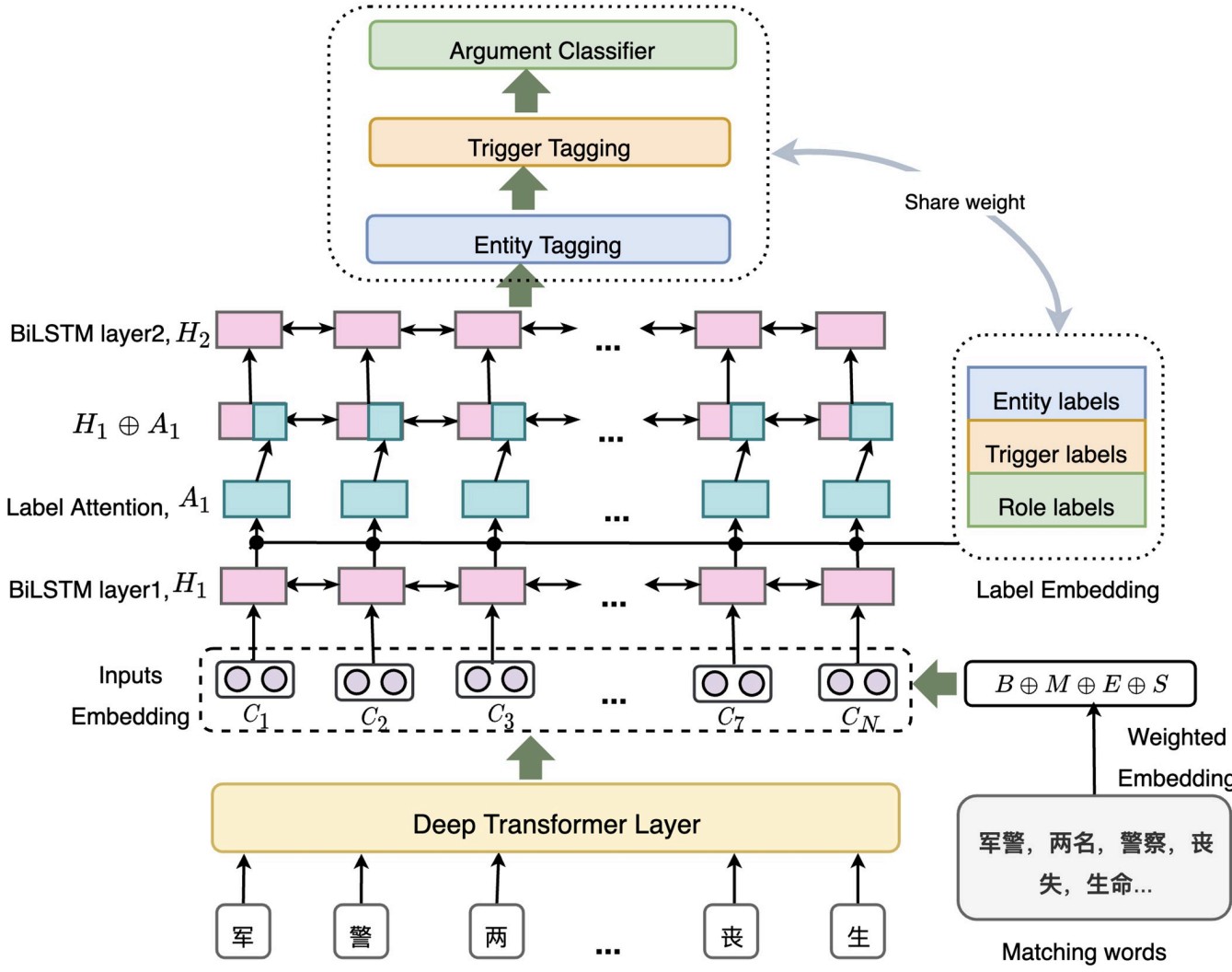

**Fig 2. Illustration of our multi-task framework for Chinese event extraction model.**

Fig 2 appears in the start of the word ("丧失") and $c_8$ ("生") appears in the end of the word. Accordingly, their corresponding segmentation categories are {**B**} and {**E**}, respectively. Note that if a segmentation label set is empty for $c_i$, we add "NULL" to the set to maintain a consistent input vector size. After categorization, words that belongs to the same segmentation label are condensed into a fixed-dimensional vector, which is formally calculated as:

$$f_i^w(K) = \frac{4}{Z}\sum_{w_i \in K} z(w_i)v^w(w_i) \tag{1}$$

where $v^w$ denotes the word embedding lookup table, $z(w_i)$ denote the frequency that a matched word $w_i$ occurs in the in-domain statistical data. $Z$ is the weight normalization term in the four segmentation sets:

$$Z = \sum_{w_i \in B \cup M \cup E \cup S} z(w_i) \tag{2}$$

In addition, we do not increase the frequency of $w_i$ if it has been counted by another sub-sequence that matches the word, thus preventing the longer words always have high frequency than the shorter words.

With Eq 2, we can obtain the combined representation of four word sets into one distributed vector as:

$$e_i^w(B, M, E, S) = [f_i^w(B) \oplus f_i^w(M) \oplus f_i^w(E) \oplus f_i^w(S)] \tag{3}$$

where $\oplus$ indicates concatenation operation.

Finally, a character $c_i$ is represented by concatenating its contextualized character embedding $e_i^c$ and its *BMES* style soft word vector $e_i^w(B, M, E, S)$ as:

$$x_i = [e_i^c \oplus e_i^w(B, M, E, S)] \tag{4}$$

## Encoder layer

After the word semantic information is incorporated, we then feed the character representations into the sequence encoding layer to capture context-sensitive features, which is implemented by stacking multi-layer bidirectional long-short term memory networks(Bi-LSTMs) [17]. This enables the preservation of the historical and future information in forward and reverse directions.

The forward and backward representations are concatenated to obtain one layer bi-directional representation of a character $i$ as $h_i = [\overrightarrow{h_i}, \overleftarrow{h_i}]$. We use matrix $H = [h_1; h_2; . . .; h_n]$ to denote stacked hidden states for input sequence $C$.

## Multi-head label attention layer

To incorporate joint label information into Bi-LSTMs, we propose to let each character's hidden representation $h_i$ to interact with all subtask labels through the multi-head attention mechanism [32].

Formally, given a set of candidate output labels $L = l_1, l_2, . . ., l_M$, we represent each label $l_m$ using an low-dimensional vector:

$$e_m^l = v^l(l_m) \tag{5}$$

where $v^l$ denotes a label embedding lookup table.

We can thus obtain label embedding matrices $E_e^l, E_t^l, E_r^l$ for entities, triggers and argument roles, respectively by feeding their one-hot categorical labels to Eq 5. $E_a^l$ is denoted as a concatenation of all label matrices $E_a^l = [E_e^l \oplus E_t^l \oplus E_r^l]$, which will then be used for calculating the label importance distribution to update input character embeddings.

Label embeddings can be randomly initialized and adjusted along with model training, or loaded more informatively with descriptive words of the label type. For example, a *BE-BORN* event is defined with coarse type "Life" and fine-grained type "Be born". Hence, for a label $l_m$, we collect all descriptive words $S_m$ and average pre-trained word embeddings in $S_m$ as the label type initial vector:

$$e_m^l = \frac{1}{|S_m|} \sum_{w \in S_m} e^w \tag{6}$$

To jointly encode features from the character subspace $h_i$ and the concatenated label subspace $E_a^l$, we apply a scaled dot-product attention:

$$u_i = h_i W^Q (E_a^l W^K) \tag{7}$$

$$a_i = \text{Softmax}\left(\frac{u_i}{\sqrt{d_h}}\right)(E_a^l W^V) \tag{8}$$

where $d_h$ is the dimension of the Bi-LSTM outputs $H$ used for forming a soft norm in the attention distribution. $W^Q$, $W^k$ and $W^v$ are model parameters.

For $m$-head attentions, we concatenate $m$ subspaces to form the final representation of $c_i$:

$$a_i = [a_i^1; ...; a_i^m] \tag{9}$$

The output of the label attention layer is the concatenation of the $i$-th step BiLSTM hidden state $h_i$ and normalized label vector $a_i$:

$$\hat{h}_i = [h_i \oplus a_i] \tag{10}$$

As illustrated in Fig 2, we then apply the second layer Bi-LSTM on top of the 0-th label-informed hidden states $\hat{H}^0 = [\hat{h}_1^0, \hat{h}_2^0, ..., \hat{h}_n^0]$ to obtain high-level $\hat{H}^1 = [\hat{h}_1^1, \hat{h}_2^1, ..., \hat{h}_n^1]$, leading to a $K$ layers hierarchical refined representations. Note that the $k$-th $\hat{H}^k$ will be fed to subsequent event decoding layer.

## Decoder layer

**Entity and trigger identification.** Given an input sequence $C$, we predict its entity tags $S^e$ and trigger tags $S^t$ by applying two feed-forward networks (FFN) with Relu activation:

$$O^e = FFN(E_e^l, \hat{H}^k) \tag{11}$$

$$O^t = FFN(E_t^l, \hat{H}^k) \tag{12}$$

where $E_e^l$ and $E_t^l$ are entity and trigger label embedding weights from section, respectively.

After that, two softmax output layers are applied to obtain the entity and trigger label probabilities:

$$P(S^e|C) = \text{Softmax}(W_e O^e + b_e) \tag{13}$$

$$P(S^t|C) = \text{Softmax}(W_t O^t + b_t) \tag{14}$$

The training objective is to minimize the negative log-probability of $\log P(S^e|C)$ and $\log P(S^t|C)$.

**Argument classification.** To obtain the argument probabilities for the entity $e_p$ with regard to the event trigger $t_q$, we combine their hidden representations from $\hat{H}^k$ and feed the concatenated vector to a softmax feed-forward layer for argument role decoding:

$$P(s_{pq}^r|c_p, c_q) = \text{Softmax}(W_r[\hat{h}_p^k; \hat{h}_q^k] + b_r) \tag{15}$$

where $\hat{h}_p^k$ and $\hat{h}_q^k$ are selected hidden states from $\hat{H}^k$ for the predicted entity $c_p$ and the trigger $c_q$ by Eq 14, $s_{ij}^r$ is the gold argument role annotation. To cope with the cases where $h_p^k$ or $h_q^k$ is a span that contains multiple consecutive tokens, we summarize their embeddings via average-

pooling for the consideration of keeping token order information. For model training, we minimize the negative the log-probability of $P(s_{pq}^r|c_p, c_q)$, which is similar to the entity and trigger classification.

**Joint training strategy.** Following the work [33], entity identification, trigger extraction and argument role classification are treated as subtasks of end-to-end event extraction, and are optimized jointly via a multi-task learning setting. A cross-entory loss is used as the object function and the log-likelihoods of all the tasks in a sentence are summarized:

$$\mathcal{L}_{joint} = \log P(S^e|C) + \log P(S^t|C) + \log P(S^t|S^e, S^t) \tag{16}$$

During the testing stage where the gold-standard entities and trigger are not available, we predict their sequence labels by choosing the output with the highest score by Eq 14 and then convert the BILOU tags to the corresponding spans and types. We next pair every entity and trigger spans to extract argument roles by Eq 15.

## Experiments

### Experimental settings

**Dataset.** To examine the effectiveness of various models on three subtasks of event extraction, we conduct experiments on a multilingual training corpus, Automatic Content Extraction (ACE) 2005 dataset [1]. The dataset contains documents mainly collected from Newswires (NW), Broadcast News (BN), Weblog (WL) fields. Following [24]'s experiment setup, we conduct tests on the Chinese dataset (ACE-CN) and the English dataset (ACE-EN). There are totally 7914 sentences in ACE-CN and 17172 in ACE-EN, respectively. We divide the training/developing/testing set accordingly. Note that we use entity types with 7 categories, event subtypes with 33 categories, and 22 argument role relations to be consistent with the pre-processing step of [24]. We follow [34] and use automatically segmented Chinese Giga-Word as the matching dictionary. Our dataset and data will be released at https://gitee.com/zjcerwin/cn_labelattn upon the paper acceptance.

**Evaluation metrics.** We use Precision (P), Recall (R) and F-Measure (F1) scores to evaluate the performances of different approaches with respect to entity recognition, event trigger detection and argument role classification by following [14, 23, 24]: **Entity**: An entity is considered correct if we can identify its start and end locations as well as the entity type correctly. **Trigger**: An event trigger is treated as correct if its start and end offsets as well as its event subtype are all matched (Trig-C). **Argument**: An argument role is determined as correct when its entity offset, relation role type and the connected triggers are all identified correctly (Arg-C).

**Pre-processing.** To represent input Chinese sentences, we use the bert-base-multilingual-cased model https://huggingface.co/transformers/pretrained_models.html for characters, and word embeddings with [35], which are pretrained on Chinese Gigaword corpus using the skip-gram model [36]. For English, we use an improved roberta-base-cased model for word pieces encoding in addition to the traditional 100-dimensional GloVe http://nlp.stanford.edu/projects/glove/ embeddings. Note that we fine-tune all the static embeddings during training and keep contextualized models fixed to keep relatively low GPU memory usage.

**Hyper-parameter settings.** All the model hyper-parameters are selected by taking the evaluation results on the developing set with the early stopping strategy. Specifically, dropout technique is adopted to prevent overfitting, which is set to 0.33 on input embeddings and hidden states. Adam optimizer is applied to adjust the network weights, with an initial learning rate of 0.01 and a decay rate of 0.85 for every five epochs. The hidden state size of stacked BiLSTM and label attention layer are employed both with 200, and the layer number is set to 2. We test the batch size in [16, 32, 64] and set the maximum epoch numbers to 150. And to

verify the superiority of the proposed method is not caused by noise in the data or other random factors, we use the pairwise t-test for measuring significance. For a fair comparison, we conduct all experiments on a machine with Intel Quad core CPU (I7-6700k, 4.0GHz) and GeForce GTX 1080 GPU with 8 GB graphic memory.

## Results

**Baselines.** With regard to prior work that considers the three subtasks, we construct baselines on word-level and character level, where word-based approaches use Jieba tokenizer https://github.com/fxsjy/jieba for segmentation, which includes:

- **Word-Tree-Joint** [12] is a typical shared-private model, which recognizes entities on top of shared Bi-LSTM representations and then extracts relations between entity pairs using tree-LSTM over dependency parsers.

- **Word-NP-pipeline** [8] is a two-stage word-based method that first pick NP nodes from a constitute parser as candidate entities, then enable triggers and arguments interactions with attention.

Character-based methods include:

- **Char-GRU-Joint** [37] is a multitask neural method considering the three subtasks by sharing Bi-GRU hidden representations.

- **Char-BERT-pipeline** [31] is a pipelined method that shares low-level BERT embeddings. To predict event mentions, we simply add a softmax transformation layer on top of the BERT encoder.

There are methods not only consider event subtasks but also involve semantic relation extraction:

- **Char-Span-Joint** [23] is a top-performed end-to-end information extraction model, all possible spans in a sentence are considered to construct information graphs.

- **Char-Global-Joint** [24] is the state-of-the-art information extraction framework that introduces indicative global features at the decoding stage to capture the cross-subtask and cross-instance interactions.

- **Transition-Joint** [27] is a recent state-of-the-art joint decoding method based on the transition system. They use a hybrid approach to incorporate character and word features [28]. For the English dataset, only word inputs are used.

To testify the effectiveness of the proposed methods, we construct two modifications:

- **Lattice**: A multi-task event extraction model with soft lexicon features replaced by lattice LSTM [28]. Note that the proposed joint label attention mechanism is not used.

- **SoftLexicon**: Using soft lexicon features and also without the label attention.

- **MLAEE + REL**: Using all techniques introduced in this work and additionally learns relation extraction with an extra FFN output layer. We use "+ REL" to indicate models involving entity relation annotations.

**Main result.** The comparison results of entity, trigger and argument extractions are shown in Table 1. We can observe that: 1) character-based methods perform better than word-based counterparts. One possible reason is that incorrect word segments would severely hurt

**Table 1. Comparison results on ACE05-CN.**

| Model | ACE05-CN | | |
|---|---|---|---|
| | Entity | Trig-C | Arg-C |
| Word-Tree-Joint | 81.2 | 58.4 | 39.5 |
| Word-NP-pipeline | 78.5 | 59.1 | 42.4 |
| Char-GRU-Joint | 83.4 | 59.6 | 45.6 |
| Char-BERT-pipeline | 87.2 | 61.6 | 45.6 |
| Char-Span-Joint* | 87.8 | 62.7 | 46.7 |
| Char-Global-Joint* | 88.5 | 65.6 | 52.0 |
| Transition-Joint* | 88.0 | 63.4 | 47.3 |
| Lattice | 87.7 | 62.4 | 50.8 |
| SoftLexicon | 88.5 | 63.3 | 51.2 |
| MLAEE | 88.6[‡] | 65.8[‡] | 54.4[‡] |
| MLAEE+REL* | **88.9**[†] | **66.4**[‡] | **55.0**[†] |

* indicates relation annotations are used.

[†] and [‡] indicate statistical significance compared to Char-Global-Joint with $p < 0.01$ and $p < 0.05$, respectively.

event results; 2) Purely character-based approaches underperform word lexicon enhanced ones **Lattice** and **SoftLexicon**, demonstrating the semantic units of Chinese words are helpful for event extraction. But instead of modifying LSTM extensively to introduce word features **Lattice**, a simplified encoding scheme **SoftLexicon** is enough and effective; 3)Compared to **Transition-Joint**, our **Lattice** give 3.5% better F-scores on argument classification. This result indicates that the joint label information is more effective than the left-to-right decoding in introducing interactive knowledge at the decoding stage, particularly for argument roles. 4) When equipped with label attention, MLAEE is 2.4% higher than current SOTA [24] on argument F-scores, verifying the effectiveness of the joint label information coupled with character representations. In addition, we evaluate the proposed framework on ACE05-EN (Table 2). The results show that MLAEE also performs well on English data, justifying the usefulness of label embedding across languages. On the other hand, there have been frameworks that jointly perform relation and event extractions [23, 24, 27]. To have a fair comparison with these models, we also integrate relation annotations into our model, denoted as "MLAEE+REL". As can be seen from Tabels 1 and 2, our MLAEE+REL can still outperform the current state-of-the-art method **Char-Global-Joint** in event trigger and argument role classification, without the

**Table 2. Comparison results on ACE05-EN.**

| Model | ACE05-EN | | |
|---|---|---|---|
| | Entity | Trig-C | Arg-C |
| Word-Tree-Joint | - | 69.6 | 50.1 |
| Char-GRU-Joint | 81.2 | 69.8 | 52.1 |
| Char-Span-Joint* | 89.7 | 69.7 | 48.8 |
| Char-Global-Joint* | **90.2** | 74.7 | 56.8 |
| Word-Transition-Joint* | 88.1 | 73.8 | 55.3 |
| MLAEE | 89.3 | 74.2 | 55.9 |
| MLAEE+REL* | 90.0 | **75.1**[†] | **56.9**[‡] |

* indicates relation annotations are used.

[†] and [‡] indicate statistical significance compared to Char-Global-Joint with $p < 0.01$ and $p < 0.05$, respectively.

**Table 3. Ablation tests on ACE-CN.**

| Settings | Trig-C | Arg-C | Entity |
|---|---|---|---|
| MLAEE | 65.8 | 54.4 | 88.6 |
| -SoftLexicon | 62.5[†] | 50.8[†] | 86.7[‡] |
| -Label embedding | 63.3[‡] | 51.2[†] | 87.3[‡] |
| -BiLSTM | 64.3[‡] | 52.6[‡] | 87.8[†] |
| -BERT embedding | 61.2[†] | 48.6[†] | 85.2[‡] |

[†] and [‡] indicate statistical significance compared to MLAEE with $p < 0.01$ and $p < 0.05$, respectively.

particular design of relation and event communications. This result further demonstrates that our label attentive model is effective across different structural prediction tasks.

**Ablation study.** To examine the influence of several key model components, we conduct ablation tests on ACE-CN. Table 3 shows the results, it can be observed that without BiLSTM, MLAEE presents a moderate drop of performances. By removing *SoftLexicon* or *Label embedding*, both trigger and argument classification degrades significantly, indicating their importance in the network. Not surprisingly, the BERT embedding brings the most performance improvements, which is consistent with the experiments in [23, 25].

**Visualization.** To understand information learned in the label embeddings, we visualize label types of entities, triggers and arguments by deducing the 200D embedding matrix into a 2D map with t-SNE after 3, 10, 40 training epochs, respectively. As shown in Fig 3, the locations of label types are increasingly more informative as training proceeds. At the initial epoch, the vectors locate randomly in the reduction space. After 10 epochs, we can observe that there are small clusters emerge, such as "Attack" event and "Attacker" argument. As training goes, we find groups start to absorb more semantic related labels, likewise "weapon", "vehicle" entity types and "victim", "target" argument roles closely surround "Attack" event. It confirms that the refined joint learning mechanism can indeed capture the label interactions among event subtasks.

**Case study.** We make a case study by comparing our **MLAEE** model with the previous best model **Char-Global-Joint**, on two representative Chinese event instances. As shown in Table 4, there are two *Attack* triggers "开" and "丢" in the first case, "枪" is the *Instrument* argument of "开" and "汽油弹" is the *Instrument* argument of "丢", respectively. It can be observed that **Char-Global-Joint** fail to identify that "丢" triggers the *Attack* event and falsely connect "汽油弹" to "开", while our **MLAEE** model can recognize two event mentions correctly. The reason is that the joint entity and event label space can incorporate correlated-type information into the network representations, leading to a positive tendency toward the recalled of event recognition. In the second case, there is much ambiguity around the phrase "向前来". The **Char-Global-Joint** yields the event trigger "向前" given that "向前" occurs more frequently than "前来" in the training set. Due to the lack of word unit semantics, it is challenging for the character-level model to infer the correct trigger and argument boundary in this case. In contrast, with the help of the soft lexicon knowledge, the **MLAEE** model detects the *Transport* trigger "前来" and *Destination* argument "日本" correctly.

## Related work

Our work mainly follows the line of event extraction and label embedding.

**English event extraction**. Identify events in English texts is a heated topic in information extraction field [3, 12–15]. Feature-based methods [4–6, 11] and recent neural-based models

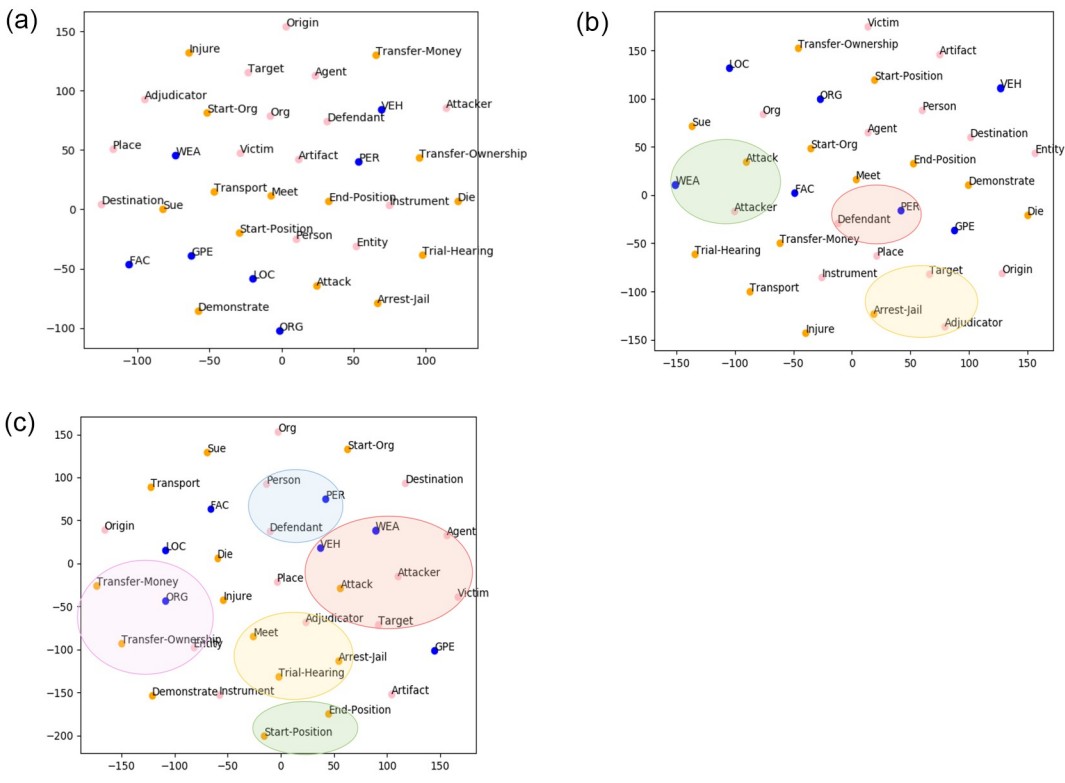

**Fig 3. t-SNE plot of joint label embeddings of entities, triggers and argument roles with varying numbers of training epochs.** (a) 3 epochs, (b) 10 epochs, and (c) 40 epochs.

[26, 37–39] have been used to promote the extraction performance continually. To improve model performance, some studies put focus on leveraging syntax information into neural networks, which include adding shortcut arcs in LSTM [8] or using Graph Convolutional Networks (GCNs) [26, 33, 37, 40, 41]. There has been work found that incorrect entity results would hurt argument role classifications significantly [9, 11, 37]. Subsequently, the transition

**Table 4. Event prediction made by different models.** Gold $C_i$ indicates the standard annotation. Words in bold and italics are correct triggers and arguments, respectively, while the underlined ones are incorrect.

| | |
|---|---|
| Gold $C_1$: | 信中明白的指出，因为被警方通缉需要钱，否则就要[开]$_{Attack}$[枪]$_{Instrument}$[丢]$_{Attack}$[汽油弹]$_{Instrument}$，让牙医师们人人自危。(The letter clearly pointed out that because being wanted by the police requires money, or else they will shoot and throw petrol bombs, putting the dentists in danger.) |
| Char-Global-Joint: | 信中明白的指出，因为被警方通缉需要钱，否则就要[开]$_{Attack}$枪丢[汽油弹]$_{Instrument}$，让牙医师们人人自危。 |
| MLAEE: | 信中明白的指出，因为被警方通缉需要钱，否则就要[开]$_{Attack}$[枪]$_{Instrument}$[丢]$_{Attack}$[汽油弹]$_{Instrument}$，让牙医师们人人自危。 |
| Gold $C_2$: | 但是会向[前来]$_{Transport}$[日本]$_{Destination}$日本的秘鲁国会调查[委员会]$_{Agent}$成员进行汇报。(However, it will report to the members of the investigation committee of the Peruvian Congress who come to Japan) |
| Char-Global-Joint: | 但是会[向前]$_{Transport}$来日本的秘鲁国会调查[委员会]$_{Agent}$成员进行汇报。 |
| MLAEE: | 但是会向[前来]$_{Transport}$[日本]$_{Destination}$日本的秘鲁国会调查[委员会]$_{Agent}$成员进行汇报。 |

based framework is devised [25] to jointly consider entity and events. Further, studies also learn relations together with events [15, 23, 24]. However, they all treat task labels as uninformative and categorical numbers. In contrast, our models map labels of event extraction tasks into semantic vectors and provide a way to realize joint learning with the interaction between the encoding and decoding stage.

**Chinese event extraction**. For Chinese, a word segmentation procedure is often required before applying event systems, despite kernel-based methods [18, 42], feature-based methods [18–20] or neural network methods [21, 24, 27, 43] are used. Instead of relying on existing segmentors, which suffer from the potential issue of error propagation we take characters as the basic units and integrate word lexicon with the input encoding scheme [28, 30].

**Label embedding**. In computer vision, research has demonstrated the importance of label embeddings, including text recognition [44] and image classification [45]. Later, work [46] shows that label embedding can benefit text classification, where text descriptions are used to generate initial label vectors. Inspired by them, [47] proposes to denoise relation instances with the help of Knowledge Graphs and entity related label embeddings. Inspired by the recent work of label attention network [48], we propose a joint label space across event subtasks, enhancing network hidden representations with global task label information. To our knowledge, we are the first to apply it for joint entity and event extraction.

## Conclusion

We present a multi-level label attentive multitask network for Chinese end-to-end event extraction. With a hierarchical refined attention mechanism, the label importance distribution is incorporated into each character's hidden state and further shares label embeddings with output layers resulting in joint learning in both the encoding and decoding stage. Results on a multi-lingual benchmark show the superiority of our model over various advanced baselines.

## Acknowledgments

We would like to thank the anonymous reviewers for their many valuable comments and suggestions.

## Author Contributions

**Formal analysis:** Wenzhi Huang.

**Project administration:** Wenzhi Huang, Junchi Zhang.

**Supervision:** Donghong Ji.

**Validation:** Junchi Zhang, Donghong Ji.

**Visualization:** Junchi Zhang.

**Writing – original draft:** Junchi Zhang.

**Writing – review & editing:** Wenzhi Huang, Junchi Zhang.

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
