## [Decision Letter · Decision Letter 0]

2 Nov 2021

PONE-D-21-29056Extracting Chinese Events with A Joint Label Space ModelPLOS ONE

Dear Dr. Zhang,

Thank you for submitting your manuscript to PLOS ONE. After careful consideration, we feel that it has merit but does not fully meet PLOS ONE’s publication criteria as it currently stands. Therefore, we invite you to submit a revised version of the manuscript that addresses the points raised during the review process.

We look forward to receiving your revised manuscript.

Kind regards,

Fu Lee Wang

Academic Editor

PLOS ONE

Journal Requirements:

Reviewers' comments:

Reviewer's Responses to Questions

**Comments to the Author**

1. Is the manuscript technically sound, and do the data support the conclusions?

Reviewer #1: Yes

Reviewer #2: Yes

Reviewer #3: Partly

2. Has the statistical analysis been performed appropriately and rigorously? 

Reviewer #1: Yes

Reviewer #2: Yes

Reviewer #3: Yes

3. Have the authors made all data underlying the findings in their manuscript fully available?

Reviewer #1: Yes

Reviewer #2: Yes

Reviewer #3: Yes

4. Is the manuscript presented in an intelligible fashion and written in standard English?

Reviewer #1: Yes

Reviewer #2: Yes

Reviewer #3: No

5. Review Comments to the Author

Reviewer #1: This paper focus on end-to-end event extraction by jointly modeling entity typing, trigger classification and argument classification. To solve the error propagation problem in Chinese event extraction and the ignorance of event type labels, this paper involves soft lexicon information, represents type labels using low-dimensional vectors and proposes label-aware attentions.

This paper is clear and sufficient, and I don't have many questions.

Here are my concerns:

1) Please provide some significant test results when comparing with other methods.

2) Which equation the line 128 refers to is not clear.

3) I think a verb is missing in the sentence of line 239 before ``3.5% better F-scores’’.

Reviewer #2: This paper proposes a joint label space framework to improve chinese event extraction, which conducts sets of experiments on a multilingual benchmark dataset. On the whole, the article is somewhat innovative and the experimental results seems to be authentic. However, I still have some concerns listed as follows:

1) for Soft Lexicon Features (equation 1-4), the matching type are easy to bring redundant errors, how to consider and solve this issue?

2) for Multi-head Label Attention Layer, adding label features have been conducted into several work, please explain the differences and innovative points.

3) for Joint Training Strategy (equation 16), do you consider the influence for entity identification, trigger extraction and argument role classification, especially for the value of the likelihood function for event triggers.

4) please explain the dividing ratio of the training/developing/testing set, how to construct the matching words？

5) why do the ablation test lack of entity performance?

6) Some grammar typos need to be corrected, especially for subject-predicate singular and plural status.

Reviewer #3: In this paper, the authors propose a Chinese event extraction system that utilizes label space information. Most traditional methods establish the event extraction pipeline without considering the underlying relationships and constraints among labels, e.g., only PERSON can DIVORCE. The authors use vectors to represent the labels and use multi-head attention to jointly train the label embedding, word embedding, sentence embedding as well as the other hidden layers. The authors also use soft lexicon features to provide additional information to mitigate the shortcoming of a character-based system. The authors demonstrate an extensive set of experiments and comparisons to prove the merit of the proposed method.

To the best of my knowledge, I think the paper needs major revision, with the following concerns:

1. The authors need to comprehensively show that their proposed method is able to solve both problems raised in the introduction. I think the current version does not completely and convincingly cover one of the motivations of the paper. In the introduction, the authors mention that Chinese texts need segmentation while the segmenter may cause errors and propagate these errors, and this argument leads the authors to use a character-based approach. However, in the experiments, although there are other baseline models with word segmentation, the authors still need an ablative setting in their proposed framework with groundtruth segmentation and system segmentation (of course these settings should remove soft lexicon features).

2. I also would like to see the examples of the proposed method, especially those examples which failed in the baselines and ablations but succeed in the proposed method.

The title of the paper indicates that the proposed framework is working on the Chinese dataset, however, I am confused that the authors show some results from the English dataset. In fact, this confusion already appears in the introductory section.

3. Another confusion is the count of ablation and modification (Line 224), the authors mention two but I see three.

4. Again in Line 224 and the following lines, I think the authors completely miss the description in REL, I guess it is relation extraction, another task. I understand any further features and tasks which are included in the joint training will boost the performance, but this definitely deviates the motivation of the proposed framework and hurts the fairness in the comparison.

5. Readers may find difficulty in reading the paper due to some informal or irregular writing and wording in the paper. Here I list a few points which the authors may consider revision:

- Introduction: Line 19~21, it seems to me that the sentence is incomplete after “thus”, and the logic of the sentence is circular.

- Line 52~57, as long as the paper is also a previous one, I suggest the authors merge this paragraph with the other potential baselines, and describe the paper in a manner that is same with the other traditional methods.

- Line 59, if the authors merge the last paragraph, logically they do not need to “contrast” the work they proposed before.

Again in Line 59, this work “proposes/introduces” etc.

- Line 123: we propose to let each character’s hidden representation hi ~~to~~ interact … (remove the “to”)

- Line 130: usually we say “updated” or “trained” when mentioning the change in the entries in the embeddings.

- Line 173: usually we name “start/end location” “offset”

- Overall, I also suggest the authors check through the whole paper, use the present tense and future tense (avoid present perfect tense and past tense), and use the active voice (e.g., “we place the embedding layer on top of”) instead of the passive voice (e.g., “the embedding layer is placed on top of”)

6. PLOS authors have the option to publish the peer review history of their article (what does this mean?). If published, this will include your full peer review and any attached files.

Reviewer #1: No

Reviewer #2: No

Reviewer #3: No

---

## [Author Response · Author response to Decision Letter 0]

26 May 2022

Reviewer #1: This paper focus on end-to-end event extraction by jointly modeling entity typing, trigger classification and argument classification. To solve the error propagation problem in Chinese event extraction and the ignorance of event type labels, this paper involves soft lexicon information, represents type labels using low-dimensional vectors and proposes label-aware attentions.

This paper is clear and sufficient, and I don't have many questions.

Here are my concerns:

1) Please provide some significant test results when comparing with other methods.

Response: Pairwise t-tests have been added to Tabel 1-3.

2) Which equation the line 128 refers to is not clear.

Response: Notation E_a^l refers to the concatenation of all label matrices. The corresponding description has been added.

3) I think a verb is missing in the sentence of line 239 before ``3.5% better F-scores’’.

Response: Addressed

Reviewer #2: This paper proposes a joint label space framework to improve chinese event extraction, which conducts sets of experiments on a multilingual benchmark dataset. On the whole, the article is somewhat innovative and the experimental results seems to be authentic. However, I still have some concerns listed as follows:

1) for Soft Lexicon Features (equation 1-4), the matching type are easy to bring redundant errors, how to consider and solve this issue?

Response: Thank you for your comment. We are sorry for the misleading description. In fact, for a character c_i, we first find all words that contain this character in a lexicon and only keep the words that can be found in the input sequence. Thus, redundant word information can be eliminated. Corresponding description has been revised in the Soft Lexicon Features section.

2) for Multi-head Label Attention Layer, adding label features have been conducted into several work, please explain the differences and innovative points.

Response: Thank you for your suggestion. Compared to the previous work[1-3] that introduce label features, our work 1)first propose a unified label embedding space for entity, event trigger and argument role extraction;2) not only enhance the network hidden representations with the global label embeddings but also share the embedding weight with the subtask output layers, thereby making full use of label knowledge.

This description has been added in the Related Work Section.

1. Wang G, Li C, Wang W, Zhang Y, Shen D, Zhang X, et al. Joint Embedding of Words and Labels for Text Classification. In: Proceedings of the 56th ACL; 2018. p. 2321–2331.

2. Hu L, Zhang L, Shi C, Nie L, Guan W, Yang C. Improving Distantly-Supervised Relation Extraction with Joint Label Embedding. In: Proceedings of EMNLP; 2019. p. 3812–3820.

3. Cui L, Li Y, Zhang Y. Label Attention Network for Structured Prediction. IEEE/ACM Transactions on Audio, Speech, and Language Processing. 2022;30:1235–1248.

3) for Joint Training Strategy (equation 16), do you consider the influence for entity identification, trigger extraction and argument role classification, especially for the value of the likelihood function for event triggers.

Response: Thank you for your suggestion. We have tried to set weighting values (\\alpha \\beta \\gamma) to control the contribution of entity identification, trigger extraction and argument role classification in the joint training. As a result, we find that set all the weighting values to one leading to the best development performance, thus simply adding all losses together. 

4) please explain the dividing ratio of the training/developing/testing set, how to construct the matching words？

Response: There are totally 7914 sentences in ACE2005 Chinese dataset, we follow[1] and divide the data into the training/developing/testing set with 6841, 526 and 547 sentences, respectively. We use automatically segmented Chinese Giga-Word as matching dictionary.

This description has been added in the Dataset Section.

[1] Yin L, et al. A Joint Neural Model for Information Extraction with Global Features, ACL 2020.

5) why do the ablation test lack of entity performance?

Response: ablation test for entity recognition has been added

6) Some grammar typos need to be corrected, especially for subject-predicate singular and plural status.

Response: addressed.

Reviewer #3: In this paper, the authors propose a Chinese event extraction system that utilizes label space information. Most traditional methods establish the event extraction pipeline without considering the underlying relationships and constraints among labels, e.g., only PERSON can DIVORCE. The authors use vectors to represent the labels and use multi-head attention to jointly train the label embedding, word embedding, sentence embedding as well as the other hidden layers. The authors also use soft lexicon features to provide additional information to mitigate the shortcoming of a character-based system. The authors demonstrate an extensive set of experiments and comparisons to prove the merit of the proposed method.

To the best of my knowledge, I think the paper needs major revision, with the following concerns:

1. The authors need to comprehensively show that their proposed method is able to solve both problems raised in the introduction. I think the current version does not completely and convincingly cover one of the motivations of the paper. In the introduction, the authors mention that Chinese texts need segmentation while the segmenter may cause errors and propagate these errors, and this argument leads the authors to use a character-based approach. However, in the experiments, although there are other baseline models with word segmentation, the authors still need an ablative setting in their proposed framework with groundtruth segmentation and system segmentation (of course these settings should remove soft lexicon features).

Response: Thank you for your valuable suggestion. Because automatic word segmentation could bring in word boundary errors, we resort to making event predictions on character-level and integrate Chinese lexicon information with soft BMES embeddings.

In this manner, our network can learn to select the most salient word features during model training, thereby avoiding potential segmentation errors. 

Unfortunately, in the ACE2005 dataset, there are no ground-truth word segmentations and we are limited in labor resources to manually label the word boundaries. Hence, it is very difficult to compare the framework with ground-truth segmentation and system segmentation.

2. I also would like to see the examples of the proposed method, especially those examples which failed in the baselines and ablations but succeed in the proposed method.

The title of the paper indicates that the proposed framework is working on the Chinese dataset, however, I am confused that the authors show some results from the English dataset. In fact, this confusion already appears in the introductory section.

Response: Thank you for your valuable suggestion. We added two representative cases in the Case Study Section. Compared to the previous best model Char-Global-Joint, our MLAEE model can fully output the correct event mention results in the cases, demonstrating the effectiveness of the proposed joint label space and the soft lexicon module.

On the other hand, in the experiment, we aim to show that our proposed joint label space model works not only for Chinese but also can be applied to other languages such as English. In addition, a strand of previous event models typically give results on English dataset. To have a fair comparison with recent advanced methods, we also show the performance of our approach on the English ACE2005 dataset.

We modified the introductory section and made a more clear description correspondingly.

3. Another confusion is the count of ablation and modification (Line 224), the authors mention two but I see three.

Response: Thank you for your suggestion. In line 224, we introduce the baseline method "Lattice-Transition-Joint" and compare it in both Chinese and English dataset. To avoid confusion with the baseline "Lattice". We have renamed "Lattice-Transition-Joint" to "Transition-Joint".

4. Again in Line 224 and the following lines, I think the authors completely miss the description in REL, I guess it is relation extraction, another task. I understand any further features and tasks which are included in the joint training will boost the performance, but this definitely deviates the motivation of the proposed framework and hurts the fairness in the comparison.

Response: Thank you for the valuable suggestion. Description of REL has been added.

The task of entity relation extraction is similar to argument role extraction.

We thus introduce relation extraction based on two considerations:1) A number of studies jointly extract relation and events [1-3]. To have a fair comparison with these models, we have constructed a modification "MLAEE + REL" (both relation and event data are used) of our "MLAEE" (only event data is used);2) We aim to demonstrate that our model can not only applied to event extraction but also entity relations. This shows its effectiveness across different structral prediction problems.

In addition, in Tabel 1 and 2, models trained relation features have been clearly marked with the tag "*". And in the Main Result section, we analysed results between models that use the same labeled data (either only events or events plus relations).

[1]Lin, Ying, et al. "A joint neural model for information extraction with global features." ACL. 2020.

[2]Wadden D , et al. Entity, Relation, and Event Extraction with Contextualized Span Representations. EMNLP; 2019.

[3]Huang W, et al. A transition-based neural framework for Chinese information extraction. Plos one. 2020.

5. Readers may find difficulty in reading the paper due to some informal or irregular writing and wording in the paper. Here I list a few points which the authors may consider revision:

Response: Thank you for such detailed suggestions. We are sorry for these informal expressions. We have proofread the paper carefully and revised the irregular writing.

- Introduction: Line 19~21, it seems to me that the sentence is incomplete after “thus”, and the logic of the sentence is circular.

Response: revised.

- Line 52~57, as long as the paper is also a previous one, I suggest the authors merge this paragraph with the other potential baselines, and describe the paper in a manner that is same with the other traditional methods.

Response: last review suggests explicitly presenting the differences between this work and our previous one. Hence, we leave the paragraph in this revision.

- Line 59, if the authors merge the last paragraph, logically they do not need to “contrast” the work they proposed before.

Again in Line 59, this work “proposes/introduces” etc.

Response: revised.

- Line 123: we propose to let each character’s hidden representation hi ~~to~~ interact … (remove the “to”)

Response: removed.

- Line 130: usually we say “updated” or “trained” when mentioning the change in the entries in the embeddings.

Response: revised.

- Line 173: usually we name “start/end location” “offset”

Response: revised.

- Overall, I also suggest the authors check through the whole paper, use the present tense and future tense (avoid present perfect tense and past tense), and use the active voice (e.g., “we place the embedding layer on top of”) instead of the passive voice (e.g., “the embedding layer is placed on top of”)

Response: the passive voice is changed to active voice.

---

## [Decision Letter · Decision Letter 1]

19 Jul 2022

Extracting Chinese Events with A Joint Label Space Model

PONE-D-21-29056R1

Dear Dr. Zhang,

We’re pleased to inform you that your manuscript has been judged scientifically suitable for publication and will be formally accepted for publication once it meets all outstanding technical requirements.

Kind regards,

Fu Lee Wang

Academic Editor

PLOS ONE

Additional Editor Comments (optional):

Reviewers' comments:

Reviewer's Responses to Questions

**Comments to the Author**

1. If the authors have adequately addressed your comments raised in a previous round of review and you feel that this manuscript is now acceptable for publication, you may indicate that here to bypass the “Comments to the Author” section, enter your conflict of interest statement in the “Confidential to Editor” section, and submit your "Accept" recommendation.

Reviewer #1: All comments have been addressed

Reviewer #2: All comments have been addressed

Reviewer #3: All comments have been addressed

2. Is the manuscript technically sound, and do the data support the conclusions?

Reviewer #1: Yes

Reviewer #2: Partly

Reviewer #3: Yes

3. Has the statistical analysis been performed appropriately and rigorously? 

Reviewer #1: Yes

Reviewer #2: Yes

Reviewer #3: Yes

4. Have the authors made all data underlying the findings in their manuscript fully available?

Reviewer #1: Yes

Reviewer #2: Yes

Reviewer #3: Yes

5. Is the manuscript presented in an intelligible fashion and written in standard English?

Reviewer #1: Yes

Reviewer #2: Yes

Reviewer #3: Yes

6. Review Comments to the Author

Reviewer #1: This paper focuses on end-to-end event extraction by jointly modeling entity typing, trigger classification, and argument classification. This paper is clear and sufficient. The authors have addressed all my previous concerns.

Reviewer #2: This paper proposes a joint label space framework to improve chinese event extraction, which conducts sets of experiments on a multilingual benchmark dataset. In summary, my comments are well considered and it can be accepted now.

Reviewer #3: I have read the manuscript again as well as the responses from the authors. I think this paper qualifies publication to the best of my knowledge.

7. PLOS authors have the option to publish the peer review history of their article (what does this mean?). If published, this will include your full peer review and any attached files.

Reviewer #1: No

Reviewer #2: No

Reviewer #3: No

---

## [Editor Report · Acceptance letter]

16 Sep 2022

PONE-D-21-29056R1 

Extracting Chinese Events with A Joint Label Space Model 

Dear Dr. Zhang:

I'm pleased to inform you that your manuscript has been deemed suitable for publication in PLOS ONE. Congratulations! Your manuscript is now with our production department. 

Kind regards, 

on behalf of

Professor Fu Lee Wang 

Academic Editor

PLOS ONE